# Oxacillin-Supplemented Mueller-Hinton Agar for In Vitro Inhibition of Ambler Class C β-Lactamases in Enterobacterales

**DOI:** 10.3390/antibiotics14060616

**Published:** 2025-06-18

**Authors:** Edgar-Costin Chelaru, Andrei-Alexandru Muntean, Mădălina-Maria Muntean, Mihai-Octav Hogea, Costin-Ștefan Caracoti, Bogdan-Florin Ciomaga, Thierry Naas, Mircea Ioan Popa

**Affiliations:** 1Discipline of Microbiology II, Department 2, Faculty of Medicine, Carol Davila University of Medicine and Pharmacy, 020021 Bucharest, Romania; edgar-costin.chelaru@drd.umfcd.ro (E.-C.C.); madalina.muntean@umfcd.ro (M.-M.M.); mihai-octav.hogea@drd.umfcd.ro (M.-O.H.); costin-stefan.caracoti@drd.umfcd.ro (C.-Ș.C.); bogdan-florin.ciomaga@drd.umfcd.ro (B.-F.C.); mircea.ioan.popa@umfcd.ro (M.I.P.); 2Cantacuzino National Military Medical Institute for Research and Development, 050096 Bucharest, Romania; 3Team Resist UMR1184 Immunology of Viral, Auto-Immune, Hematological and Bacterial Diseases (IMVA-HB), INSERM, Faculty of Medicine, CEA, LabEx LERMIT, Université Paris-Saclay, 94270 Le Kremlin-Bicêtre, France; thierry.naas@aphp.fr; 4Associated French National Reference Center for Antibiotic Resistance: Carbapenemase-Producing Enterobacteriaceae, 94270 Le Kremlin-Bicêtre, France; 5Bacteriology-Hygiene Unit, Bicêtre Hospital, Assistance Publique-Hôpitaux de Paris (AP-HP), 94270 Le Kremlin-Bicêtre, France

**Keywords:** oxacillin, cloxacillin, AmpC, Enterobacterales, Mueller-Hinton agar, disk diffusion antibiogram

## Abstract

**Background**: The increasing incidence of infection with Gram-negative bacilli (GNB) producing broad-spectrum β-lactamases, such as extended-spectrum β-lactamases (ESBLs), cephalosporinases (AmpCs), and carbapenemases, has become a great clinical concern. AmpCs are found in many clinically relevant Enterobacterales, where they may compromise the effectiveness of most β-lactams, including carbapenems when associated with an impaired outer membrane. Detection and distinction between these resistance mechanisms are crucial for antimicrobial therapy and for implementation of proper infection control procedures to prevent further spread. **Methods**: The disk diffusion antibiogram using Mueller-Hinton agar (MHA) supplemented with cloxacillin (MHC), which inhibits AmpCs, was validated to identify AmpC-producing Enterobacterales (AmpC-PE). As cloxacillin is not available in several countries, we investigated the use of oxacillin as an alternative compound to inhibit AmpCs. The ability of MHA supplemented with oxacillin (MHO) to distinguish between carbapenem-resistant Enterobacterales (CREs) due to AmpC hyperproduction and the presence of a carbapenemase has particularly been investigated. **Results**: MHOs containing several concentrations of oxacillin were compared to MHA and MHC containing 250 mg/L cloxacillin (MHC250). A set of well-characterized Enterobacterales with different β-lactam resistance mechanisms were evaluated. MHO containing 300 mg/L of oxacillin (MHO300) gave similar results to MHC250. **Conclusions**: The use of MHO300 proved to be efficient in inhibiting AmpCs, allowing differentiation between AmpC hyperproducers and carbapenemase producers. In addition, the use of MHO300 allowed detection of resistance mechanisms hidden by AmpCs, such as ESBLs.

## 1. Introduction

Modern medicine relies on the use of antibiotics, including β-lactams, which, due to their safety, reliable killing properties, and clinical effectiveness, are among the most frequently prescribed antibiotics to treat bacterial infections. However, their utility is threatened by the global proliferation of β-lactamases with broad hydrolytic capacities, particularly in multidrug-resistant (MDR) Gram-negative bacteria (GNB) [1]. β-lactamases are divided into four classes based on their sequence identities, with classes A (penicillinase), C (cephalosporinase), and D (oxacillinase) being active-site serine enzymes, while class B represents metallo-β-lactamases (MBLs) [2]. β-lactamase-mediated resistance extends towards carbapenems, whose activity is challenged by true carbapenemases (classes A or D) [2], but overproduction of cephalosporinases (also known as AmpC enzymes) or ESBLs, which weakly hydrolyze carbapenems, combined with reduced porin expression (penetration defect) may also lead to resistance to carbapenems [2].

AmpCs may be chromosomally encoded and naturally produced in several Enterobacterales (e.g., in *Enterobacter* spp., *Citrobacter* spp., *Serratia* spp.). In these species, the expression may be induced in the presence of sub-inhibitory concentrations of inducing β-lactam antibiotics, such as cefoxitin or carbapenems. A constitutively high-level of expression may be achieved by mutations in genes involved in the AmpC expression, such as AmpD or AmpR, or by mobilization of the *bla*_AmpC_ gene onto plasmids. AmpC-hyperproducing Enterobacterales (AHEs), especially in conjunction with ESBL production and other non-enzymatic mechanisms (impaired outer membrane permeability, active efflux, mutations in PBPs), may be resistant to broad spectrum β-lactams, including carbapenems. Being able to distinguish between these mechanisms is crucial for antimicrobial stewardship and infection control [3,4,5,6]. In addition, as AmpCs are not inhibited by clavulanic acid, their overexpression, leading to expanded-spectrum cephalosporin resistance, may hide other underlying β-lactam resistance mechanisms, such as ESBL production [3,4].

Resistance to certain antibiotics and double-disk synergy images on a disk diffusion antibiogram may be useful to identify ESBLs, AmpCs, and carbapenemases [7,8,9,10,11,12]. Although EUCAST recommends testing for AmpC in Enterobacterales with a cefoxitin (FOX) diameter of inhibition < 19 mm (MIC > 8 mg/L) and resistance to ceftazidime and/or cefotaxime, ACC-type AmpCs, naturally produced by *Hafnia alvei* or plasmid encoded in Enterobacterales, are not detected, as they do not hydrolyze cefoxitin [4]. Similarly, cefoxitin resistance may also be the result of porin loss [13]. Thus, the current recommendations include testing the inhibition property of AmpC with cloxacillin or boronic acid derivatives; the latter’s drawback is that it also inhibits Ambler class A enzymes. Finally, molecular methods, like PCR or whole-genome sequencing (WGS), which are more expensive, laborious, and less available, are the most accurate solutions to detect a gene, but not its expression [4,14,15,16].

Cloxacillin, an antibiotic classically used to treat infections with Gram-positive cocci, is not active on Gram-negative bacteria at doses compatible with clinical use due to impermeability issues. Nevertheless, it has been shown to inhibit AmpC enzymes in vitro, and in bacteria at concentrations way above clinical dosage [17,18]. Thus, cloxacillin has been added in various testing methods to confirm the presence of AmpCs by restoring susceptibility to extended-spectrum cephalosporins (ESCs). In addition, by inhibiting AmpCs, hidden resistance mechanisms to ESCs (such as ESBLs) and to carbapenems (such as carbapenemases) can be uncovered and observed phenotypically (e.g., third-gen. cephalosporin—clavulanic acid synergy). Thus, in Enterobacterales, it is recommended to use Mueller-Hinton agar (MHA) supplemented with 200–250 mg/L of cloxacillin (MHC250). An increase of at least 5 mm of the inhibition diameters for β-lactams, classically hydrolyzed by AmpCs on MHC250 as compared to MHA after incubation (e.g., 16 h at 37 °C), confirms the presence of a cephalosporinase [3,8,9,10,11,14,19,20,21,22,23,24,25,26,27,28,29,30].

Although oxacillin has been described as a cephalosporinase inhibitor, its routine use has not been documented in the literature [3]. As oxacillin is currently available in countries where cloxacillin is not (e.g., some Eastern European countries, including Romania and those of the Balkans), this study aimed to validate the cephalosporinase inhibitor potential of oxacillin and to present a phenotypic method for the differential diagnosis of AmpC producers (associated or not with other resistance mechanisms) using Mueller-Hinton supplemented with oxacillin (MHO). The AmpC inhibition properties of oxacillin were evaluated on a variety of β-lactamase-producing Enterobacterales, and the oxacillin concentration that gives similar test performances, as compared to the most widely used cloxacillin concentration, e.g., 250 mg/L, was determined [9,10,31,32].

## 2. Results

### 2.1. Oxacillin Concentration Determination

Using a panel of 16 CASE (cephalosporinase) and HCASE (hyperproduced cephalosporinase) producing Enterobacterales, similar inhibition results were obtained with MHO300 and 350 mg/L as compared to the classical MHC250 (Figure 1). Individual results for all the bacteria and antibiotics tested on MHA, MHC250, and MHO300 are presented in Appendix A.

Further testing of all the 113 strains using MHO300 revealed similar results to MHC250. An excessive increase in diameters was observed for several strains when evaluated individually on MHO supplemented with 350 mg/L, with diameters increased by over 5 mm around the ETP disk for 11 strains: 9 strains of *Enterobacter cloacae* harboring carbapenemases (5 IMI-, 1 NmcA, 1 IMP-8, 1 VIM-1, 1 VIM-4) and 2 strains of *Klebsiella pneumoniae*, 1 harboring IMP-8 and the other ESBL. On MHO supplemented with 300 mg/L oxacillin, for 4 of these 11 strains (1 VIM-4 *E. cloacae*, 1 IMI-17 *E. cloacae* and the 2 strains of *K. pneumoniae*), the diameter around ETP did not increase over the threshold, while the overall results were better and closer to MHC250. The remaining seven strains of *E. cloacae* still presented increased diameters on both MHC250 and MHO300 around the ETP disk. In addition to these strains, *E. coli* ATCC 25922 often failed to grow on MHO350.

The differences between the inhibition diameters obtained on each culture medium (MHA, MHC250, and MHO300) were evaluated using the Kruskal–Wallis Test. Statistically significant differences were observed between the three media with the exception of ticarcillin (TIC), *p* = 1.358, and noting that for aztreonam (ATM) the difference trends toward statistical significance but does not reach *p* < 0.05. As the Kruskal–Wallis test showed significant differences between the three groups, the difference between the diameters obtained from the paired samples on the media containing inhibitors (MHC250 and MHO300) was further evaluated using the Wilcoxon signed-rank test, which showed no statistical significant differences between the two media (Table 1).

Aggregate analysis showed no significant difference between MHO300 and MHC250 for each of the seven antibiotics tested, as seen in Figure 2 (ticarcillin not included). Thus, the MHO300 medium proved to be efficient in differentiating between cephalosporinase producers and non-producers.

To evaluate the antibiotic disks, the culture media and the impact of the concentration of 300 mg/L of oxacillin on the growth of *E. coli* ATCC 25922, a disk diffusion antibiogram was performed on MHA, MHC250, and MHO300. Overall, the inhibition diameters were identical on the antibiotic-supplemented media, and no significant increase in diameters was noticed when compared to MHA, as it presents a natural, weakly expressed cephalosporinase. The growth was fainter on MHC250 and MHO300, which was also probably due to its natural CASE and inherent susceptibility (Appendix A), sometimes hardly even visible. In addition to *E. coli* ATCC 25922, two other isolates completely failed to grow on the selective media. *E. coli* harboring plasmidic DHA-1 repeatedly failed to grow on both MHO300 and MHC250 (Appendix A) and an AmpC-hyperproducing *H. alvei* isolate gave faint growth on MHC250 and no growth (except for a few colonies) on MHO300 (Appendix A).

### 2.2. Penicillinase and ESBL Producers

All three class A penicillinase producers without AmpC displayed large inhibition diameters, with no significant increase in diameters on MHC250 or MHO300 (Appendix A). The Enterobacterales producing OXA-type penicillinases (n = 2) revealed similar inhibition diameters on all the tested media (Figure 3A). All 29 ESBL (class A) producers with no AmpC, including those that associate with other resistance mechanisms conferring overall increased β-lactam resistance, showed no significant difference in inhibition diameters (Figure 3B,C).

### 2.3. CASE and HCASE Producers

Thirty-eight (38) cephalosporinase producers were evaluated. Similarly to *E. coli* ATCC 25922, a strain of *E. coli* encoding for a weakly expressed CASE presented insignificant increase in diameters, and a non-inducible and basal CASE-producing *Enterobacter cloacae* revealed increased inhibition diameters of a maximum of 4 mm. Two other isolates, an HCASE + ESBL-producing *E. cloacae* and a plasmid-encoded CASE + ESBL-producing *K. pneumoniae* that displayed MICs for ertapenem > 32 mg/L, revealed only a slight increase in diameters (2–3 mm) on both MHC250 and MHO300. However, on both antibiotic-containing plates, a synergy image between CAZ (ceftazidime) and/or CTX (cefotaxime) and TCC (ticarcillin/clavulanic acid) was evidenced, suggesting the presence of an ESBL (Figure 4A).

Identification of ESBLs in AmpC producers is often difficult because the hyperproduction of AMPCs may hide the ESBL phenotype. This was the case for 5/12 isolates co-producing an AmpC and an ESBL, for which no synergy image could be evidenced between third-generation cephalosporins (CTX and/or CAZ) and clavulanic acid (TCC disk) on MHA. The use of MHO300 and MHC250 on these five isolates resulted in slightly increased diameters around CTX, with clear synergy images between CTX and TCC, suggesting the presence of an ESBL (Figure 4B).

Of the remaining 34 isolates, 33 exhibited differences in inhibition diameters of at least 5 mm on MHO300 and 32 on MHC250, as compared to MHA (Figure 5A–C and Appendix A). The 1 and 2, respectively, out of the 34 strains for which the diameters did not increase over 5 mm were as follows: 1 HCASE + ESBL-producing *E. cloacae*, for which an MIC of 12 mg/L for ertapenem was defined and no increase in diameters was observed; and 1 CASE + ESBL-producing *E. cloacae*, for which some diameters increased over the threshold on MHO, but not on MHC250. For some strains, the AmpC inhibition can lead to very large inhibition areas (*E. cloacae* harboring IMI-1 carbapenemase and inducible chromosomal AmpC). This behavior was observed in other IMI-/NmcA-producing *Enterobacter* spp. isolates too (Appendix A).

Sometimes, mutants continue to appear in the inhibition zone for AmpC-producing strains. For the purposes of diagnosing AmpC hyperproduction, they should be ignored, and the outer diameter measured (Figure 5D).

### 2.4. Carbapenemase Producers

Of the carbapenemase-producing isolates, including those with low carbapenemase activity, 30/41 displayed no significant increase in diameters either on MHC250 or on MHO300 (Figure 6A–D). The 11 strains for which a diameter increase of at least 5 mm was observed on both MHC and MHO were as follows: 1 VIM-1 *E. cloacae* (only the diameter around ETP was increased); 1 NDM-1 *Providencia stuartii* (around other disks than ETP); 1 KPC-2 *E. cloacae* (around other disks than ETP); 1 VIM-4 *E. cloacae* (only the diameter around ATM was increased); 7 IMI- or NmcA-producing *E. cloacae* (for two strains around disks other than ETP).

## 3. Discussion

Identification of ESBLs in AmpC producers is often difficult because the hyperproduction of AmpCs may cover the phenotype resulting from the ESBL. Yet with hyper-AmpC-producing Enterobacterales standard infection control precautions are enough to limit their spread, but with ESBL producers additional contact precautions are necessary. Similarly, with plasmid-encoded carbapenemase producers, the risk of spread is much higher than with ESBL or AmpC producers due to the fact that outer membrane impermeability defects may also lead to carbapenem resistance [33,34]. In addition, with novel β-lactams and β-lactam/inhibitor combinations, knowing the underlying resistance mechanisms is mandatory to use them efficiently [35]. The use of cloxacillin-containing plates has now widely been adopted to inhibit AmpCs to reveal hidden resistance mechanisms [3,4]. However, cloxacillin-containing plates are often too expensive for resource-limited countries to purchase. In addition, cloxacillin is not available in every country, so even preparing in-house plates may be impossible. In these countries, oxacillin is used instead. Here, we evaluated the use of a commercial and injectable form of oxacillin as an alternative to cloxacillin for efficient inhibition of AmpCs.

As observed for cloxacillin-containing plates, some Enterobacterales are highly susceptible to oxacillin, resulting in no growth on these plates [8]. Previously, authors have reported different concentrations of MHC, ranging from 200 to 250 mg/L or more [7,9,10,36]. One could consider a similar strategy for MHO, as small differences in concentration (50 mg/L) could significantly influence the diameters. Although for many strains the results were still acceptable on MHO350 when compared to MHC250, in some cases diameters increased > 5 mm for some non-AmpC CPE, up to the absence of growth for some IMI-/NmcA-producing *E. cloacae.* On MHO300, the results were, however, similar to those of MHC250; thus MHO300 was retained.

Impaired growth on oxacillin and cloxacillin plates was observed with 3 out of the 113 isolates tested. This phenomenon has previously been observed for some wild-type or antibiotic-susceptible isolates on MHC250 [8]; faint growth was also observed for ATCC 25922 on both MHC250 and MHO300 (Appendix A), sometimes even resulting in absence of growth. *E. coli* harboring plasmidic DHA-1 failed to grow on both MHO300 and MHC250. In fact, the only phenotypic indication of β-lactam resistance was that to cefoxitin (Appendix A). For an AmpC-hyperproducing *H. alvei*, faint growth appeared on MHC250 and no growth (with the exception of a few colonies) on MHO300 (Appendix A). For one *E. cloacae* producing NmcA, faint growth on both MHC250 and MHO300 was observed, but with continuous culture only around the cefoxitin (FOX) disk, which displayed the highest resistance on MHA (Appendix A). This phenomenon might be caused by the inhibition of the natural cephalosporinase that some bacteria harbor. This leads to increased susceptibility to the antibiotic disks (leading to very large diameters) or, as mentioned earlier, they might actually show a high susceptibility to cloxacillin and/or oxacillin which, infused in high concentrations into the medium, not only inhibits the AmpC, but makes it difficult for the tested bacteria to grow. One possible solution is, as discussed above, to consider using different concentrations of oxacillin (lower in this situation). However, it must be mentioned that strains that show this behavior are actually very susceptible to β-lactam antibiotics, and the screening or testing for the presence of AmpC would not be applied routinely for them.

For some AmpC producers the absence of or insufficient increase in diameters (1/34 on MHO300 and 2/34 on MHC250) was observed. This is likely due to the association of other resistance mechanisms (including efflux pumps and porin loss) and/or higher MICs such as with one hyperproduced cephalosporinase + ESBL *E. cloacae*, with ETP MIC = 12 mg/L. The same explanation might apply for the two strains (one hyperproduced cephalosporinase + broad-spectrum penicillinase + ESBL-producing *E. cloacae* and one hyperproduced cephalosporinase + ESBL-producing *K. pneumoniae* with an ertapenem MIC of over 32 mg/L), for which the diameters did not increase over the threshold, but the ESBL underneath was revealed after the AmpC inhibition. One cephalosporinase + ESBL *E. cloacae* for which some diameters increased over the threshold on MHO300, but not on MHC250, was probably due to the low expression of the cephalosporinase.

It must be emphasized that sometimes, slight diameter increases (<5 mm) can be observed in non-AmpC producers and in microorganisms presenting natural cephalosporinases (such as *Escherichia* or *Enterobacter*). As such, correct measurement of the diameters and interpretation of the results on MHC and MHO in the context of natural resistances and of the disk diffusion antibiogram results (diameter around “key antimicrobials” such as cefoxitin, carbapenems, aztreonam, ticarcillin, ceftazidime-avibactam, etc.) must be considered.

It must be noted that 11 CPEs presented an increase of over 5 mm on MHC250 and MHO300, but 10 of these are *E. cloacae* (natural inducible AmpC). The results may be due to specific (e.g., the presence of a natural, inducible cephalosporinase in strains pertaining to *Enterobacter* spp.) or non-specific reasons. The diameter around ETP was increased over the threshold (5 mm) for one VIM-1 *E. cloacae*, and the diameters around other antibiotics for one NDM-1 *P. stuartii* (probably due to a low-expression NDM-1) and for one KPC-2 E. cloacae. The diameter increase around ATM for one VIM-4 *E. cloacae* could be explained by the inhibition of its natural AmpC, revealing the characteristic susceptibility of metallo-β-lactamases producers to ATM. However, it is important to highlight that, although these enzymes are rare, for seven/eight IMI- or NmcA-producing *E. cloacae*, the diameters increased around most of the tested antibiotics (for two strains around disks other than ETP), on both MHC250 and MHO300. These results could be explained by the inhibition of the inducible, chromosomal natural AmpC present in *Enterobacter* spp. and the reduced enzymatic activity of IMI-/NmcA carbapenemases.

Although culture media supplemented with oxacillin could be an alternative to cloxacillin-supplemented media and a cheap way of detecting and/or inhibiting AmpC in vitro, not all laboratories have the logistic capacity to produce in-house culture media, and special approval must be obtained to produce these culture media. In addition, constant availability of large quantities of oxacillin could be a limiting factor, making it possible only for certain units to produce large quantities of such media. Although 300 mg/L of oxacillin is the optimal concentration for Enterobacterales, the concentration has to be evaluated on non-fermenters (*Pseudomonas* spp. and *Acinetobacter* spp.), but it is expected that higher concentrations are needed, as illustrated with cloxacillin and these bacteria [24,25,26,27,28,29].

## 4. Materials and Methods

### 4.1. Media Preparation

Three types of culture media were used: Mueller-Hinton agar (MHA, Oxoid, Basingstoke, UK), an in-house prepared MHA supplemented with 250 mg/L cloxacillin (MHC250), using a dehydrated MHA base (Oxoid, Basingstoke, UK) and cloxacillin powder (Orbenin^®^ 1g injectable powder, Astellas Pharma, Levallois Perret, France), and an in-house MHA (dehydrated base, Oxoid, Basingstoke, UK) supplemented with various concentrations of oxacillin (Oxacilina^®^ 500 mg/1000 mg injectable powder, Antibiotice, Iași, Romania), ranging from 250 mg/L to 400 mg/L (MHO250–MHO400).

The media were prepared according to the producer’s instructions, and the antibiotic solutions (cloxacillin or oxacillin powder, previously sterilized by the manufacturer and packaged in ampules resuspended in distilled water sterilized by autoclaving) were aseptically added using an Eppendorf Research Plus 100–1000 μL pipette and sterile filter tips (Eppendorf, Hamburg, Germany) before pouring the medium into sterile Petri dishes, after the medium had reached a temperature of 50–55 °C. The media were poured into 90 mm diameter Petri dishes (FL MEDICAL s.r.l. Unipersonale, Torreglia, Italy) in order to obtain a thickness of 4 mm (+/−0.5 mm). The plates were stored at 4 °C prior to use, for at least one day.

### 4.2. Susceptibility Testing

Inocula of 0.5 McFarland were prepared in saline water (Cantacuzino Institute, Bucharest, Romania) and subsequently plated using sterile swabs (Aptaca S.p.A., Canelli, Italy) on MHA, MHC, and MHO plates, following the EUCAST guidelines [37]. The plates were incubated for 16 h at 37 °C, and the diameters were measured by two investigators using a ruler or caliper.

The antibiotic discs used were ertapenem (ETP) 10 μg, aztreonam (ATM) 30 μg, cefoxitin (FOX) 30 μg, ticarcillin (TIC) 75 μg, ceftazidime (CAZ) 10 μg, cefotaxime (CTX) 5 μg, and ticarcillin/clavulanate (TCC) 75/10 μg (Oxoid, Basingstoke, UK) [38].

As inhibition diameters, ≥5 mm on MHC compared to MHA, were shown previously to confirm the presence of an AmpC [9,10,11,14,22], the same 5 mm threshold value was applied for MHO to be considered positive for AmpC.

The CAZ and CTX disks were placed close to the TCC disk (15 mm) to detect the production of an ESBL, especially in AmpC producers. The resulting diameters on MHA and MHC250 were compared to those on MHO. This allows the differential evaluation of individual antibiotic diameters under standard conditions, when supplemented with oxacillin/cloxacillin, as well as when performing a double-disk synergy test (DDST) [4,7,39].

The optimal oxacillin concentration was determined on a subset of isolates listed in Table 2. MHO media supplemented with 250, 300, 350, and 400 mg/L were compared to MHC250 and MHA. The oxacillin concentration that gave similar results to MHC250 was retained for further study.

### 4.3. Bacterial Isolates Tested

A total of 113 well-characterized Enterobacterales strains with various β-lactam resistance mechanisms were used in the study (Table 2). They included Enterobacterales producing cephalosporinases (CASE; n = 8), of which 4 were co-producing an ESBL (3 class A and 1 class D); hyperproducing cephalosporinases (HCASE; n = 30), of which 9 were co-producing a class A ESBL; producing carbapenemases (CPEs; n = 41); ESBLs (n = 31); and penicillinases (n = 3). A wild-type *Escherichia coli* ATCC 25922 strain was used as a control for the antibiotics and the culture media used. The CASE strains usually presented a reduced diameter around a cefoxitin disk (FOX), while the HCASE strains also showed a reduced diameter around 3rd-generation cephalosporins [ceftazidime (CAZ) and cefotaxime (CTX)].

The isolates were subcultured from frozen stocks (stored at −80 °C) on UriSelect 4 (BioRad, Marne-la-Coquette, France) and incubated for 18–24 h at 37 °C, then on Mueller-Hinton agar (Oxoid, Basingstoke, UK) for an additional 18–24 h at 37 °C. The cultures obtained were used to prepare the suspensions for susceptibility testing.

### 4.4. Statistical Analysis

The overall concordance of the MHC-MHO inhibition zone diameters was analyzed using the R statistical software (version 4.3.3); the ANOVA test was performed for the in-between group comparison and then Tukey’s HSD for pos hoc analysis. Also, data corresponding to the inhibition diameters were analyzed using the Kruskal–Wallis test and the Wilcoxon signed-rank test, the latter being a special test for paired samples, showing the differences between the two groups by comparing the diameter differences obtained for each strain on both inhibitor-supplemented culture media.

## 5. Conclusions

Use of Mueller-Hinton agar supplemented with oxacillin in a concentration of 300 mg/L is a cheap and effective way to identify cephalosporinase producers, to differentiate AmpC hyperproducers from non-AmpC producers and CPEs, and to reveal AmpC-masked resistance mechanisms, such as ESBLs. Oxacillin might be an interesting alternative, especially in countries where cloxacillin is not available.

The possibility to locally produce a culture medium such as MHO 300 mg/L can be helpful in avoiding prolonged delivery times and inadequate storage and transportation conditions, risks that may occur in the case of imported, ready-to-use products, such as MHA supplemented with 250 mg/L of cloxacillin.

The obtained results on oxacillin may be further adapted for other tests using cloxacillin as a cephalosporinase inhibitor, such as Rosco Diagnostica KPC and the MBL confirm kit (RDCK™), cefotetan-cloxacillin E-test strips, rCIM-A, or the modified Hodge test using Mueller-Hinton with cloxacillin [8,9,10,21,22,40].

## Figures and Tables

**Figure 1 antibiotics-14-00616-f001:**
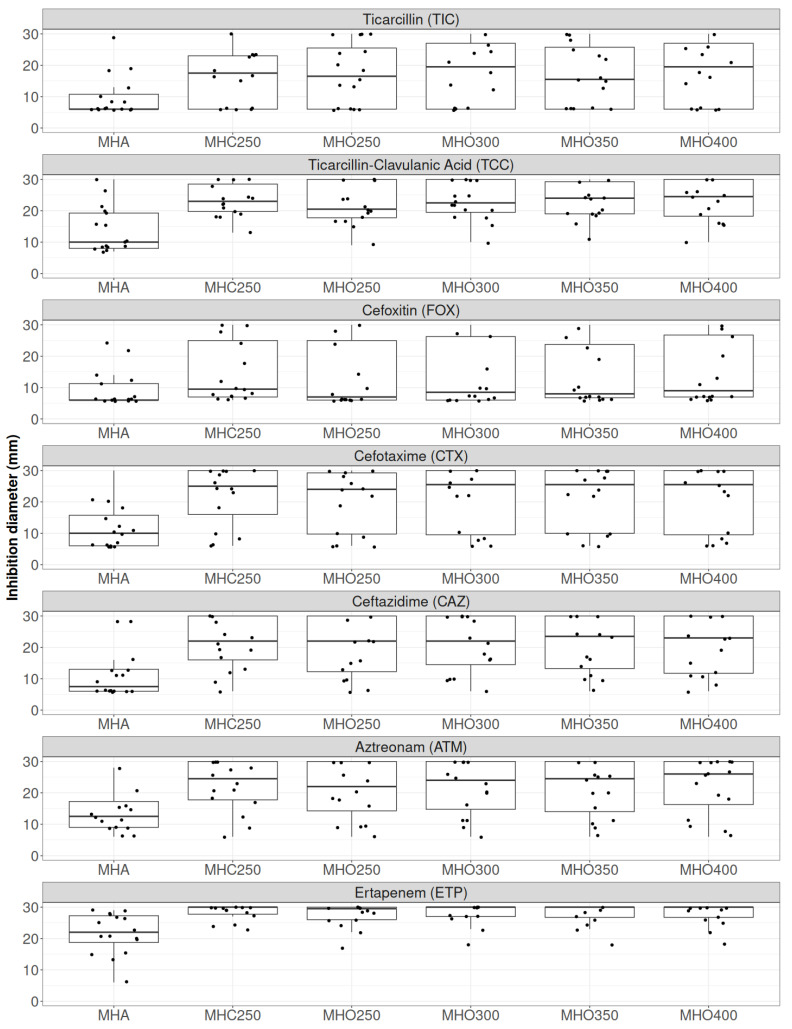
Comparison of inhibition diameters on the different media tested using the 16 AmpC-producing Enterobacterales and the 7 tested antibiotics [ATM: aztreonam (30 μg); CAZ: ceftazidime (10 μg); CTX: cefotaxime (5 μg); ETP: ertapenem (10 μg); FOX: cefoxitin (30 μg); TIC: ticarcillin (75 μg); TCC: ticarcillin/clavulanic acid (75/10 μg)]. MHA: Mueller-Hinton agar; MHC250: Mueller-Hinton agar supplemented with 250 mg/L of cloxacillin, and MHO250/300/350/400 supplemented with 250/300/350/400 mg/L of oxacillin, respectively. Each data point represents the individual strains tested.

**Figure 2 antibiotics-14-00616-f002:**
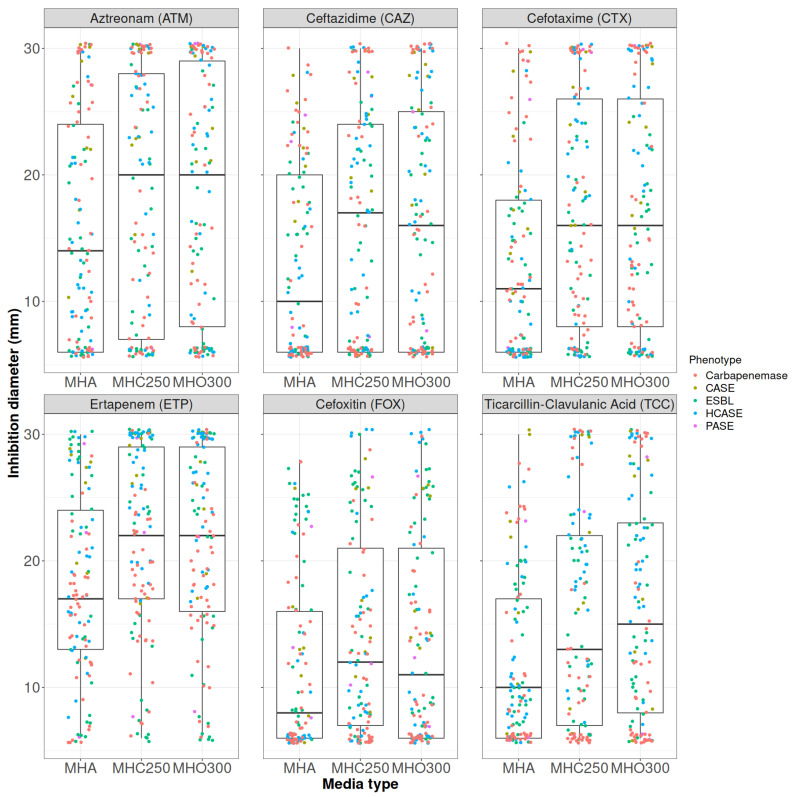
Boxplot showing significant differences between MHA and MHO300 (similar to the differences between MHA and MHC250) of at least 5 mm for cephalosporinase producers. No significant difference was observed between MHO300 and MHC250 for ATM: aztreonam (30 μg); CAZ: ceftazidime (10 μg); CTX: cefotaxime (5 μg); ETP: ertapenem (10 μg); FOX: cefoxitin (10 μg); and TCC: (ticarcillin/clavulanic acid: 75/10 μg). MHA: Mueller-Hinton agar; MHC250: Mueller-Hinton agar supplemented with 250 mg/L cloxacillin; MHO300: Mueller-Hinton agar supplemented with 300 mg/L oxacillin. Each data point represents the individual strains tested. The strains were grouped based on their resistance phenotype: CASE = cephalosporinase; ESBL = extended-spectrum β-lactamase; HCASE = hyperproduced cephalosporinase; PASE = penicillinase.

**Figure 3 antibiotics-14-00616-f003:**
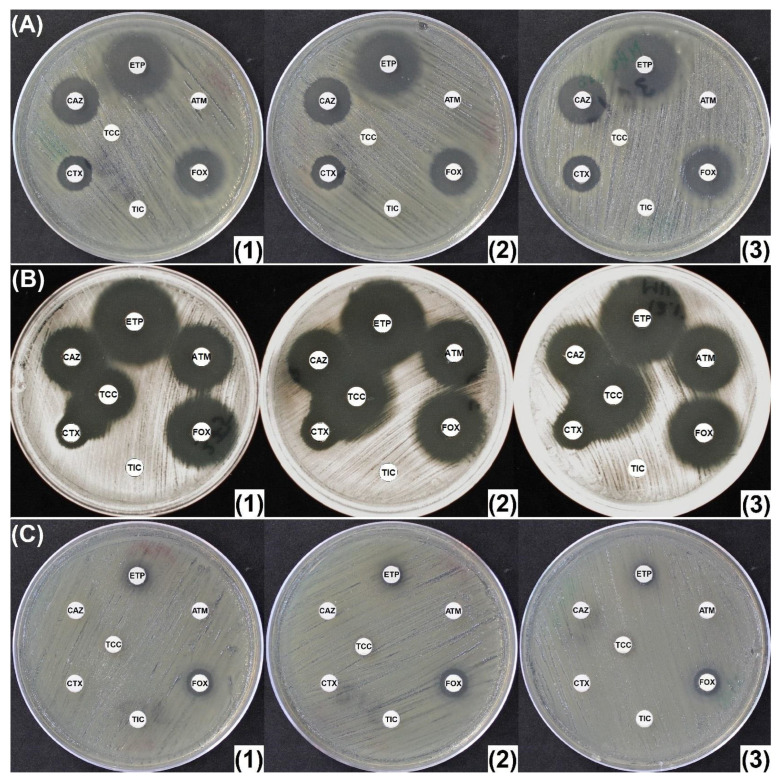
Class A and D β-lactamase-producing Enterobacterales: (**A**) OXA-2-like (class D ESBL)-producing *Klebsiella oxytoca*; (**B**) ESBL (CTX-M-14)-producing *Klebsiella pneumoniae*; (**C**) Carbapenem-resistant ESBL-producing *K. pneumoniae*. (**1**) Mueller-Hinton agar (MHA); (**2**) Mueller-Hinton agar supplemented with 250 mg/L cloxacillin (MHC250); (**3**) Mueller-Hinton agar supplemented with 300 mg/L oxacillin (MHO300). ATM: aztreonam (30 μg); CAZ: ceftazidime (10 μg); CTX: cefotaxime (5 μg); ETP: ertapenem (10 μg); FOX: cefoxitin (30 μg); TIC: ticarcillin (75 μg); TCC: ticarcillin/clavulanic acid (75/10 μg).

**Figure 4 antibiotics-14-00616-f004:**
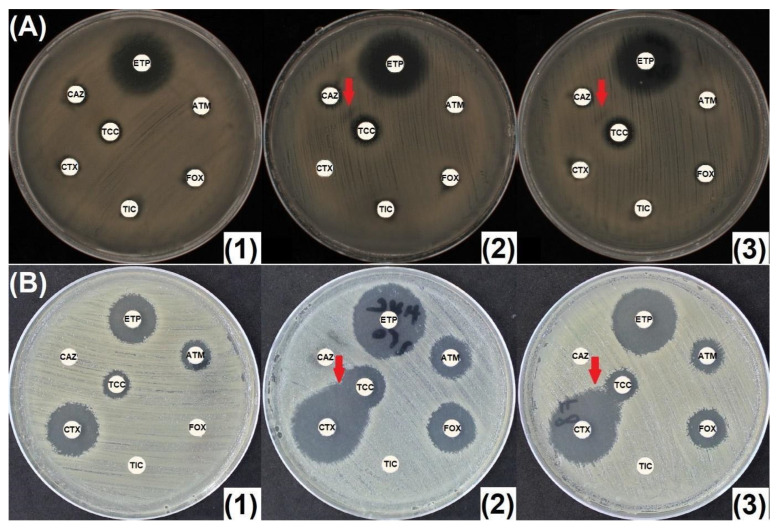
(**A**) *K. pneumoniae* co-harboring hyperproduced cephalosporinase and ESBL with an MIC of over 32 mg/L for ertapenem. AmpC inhibition on MHC250 and MHO300 leads to a slight uncovering of the ESBL underneath; (**B**) cephalosporinase- and ESBL (TEM-24)-producing *Enterobacter aerogenes*. The increased diameters together with cefotaxime (CTX)—ticarcillin/clavulanate (TCC) synergy (red arrow) confirm the presence of a cephalosporinase and an ESBL. (**1**) Mueller-Hinton agar (MHA); (**2**) Mueller-Hinton agar supplemented with 250 mg/L cloxacillin (MHC250); (**3**) Mueller-Hinton agar supplemented with 300 mg/L oxacillin (MHO300). ATM: aztreonam (30 μg); CAZ: ceftazidime (10 μg); CTX: cefotaxime (5 μg); ETP: ertapenem (10 μg); FOX: cefoxitin (30 μg); TIC: ticarcillin (75 μg); TCC: ticarcillin/clavulanic acid (75/10 μg).

**Figure 5 antibiotics-14-00616-f005:**
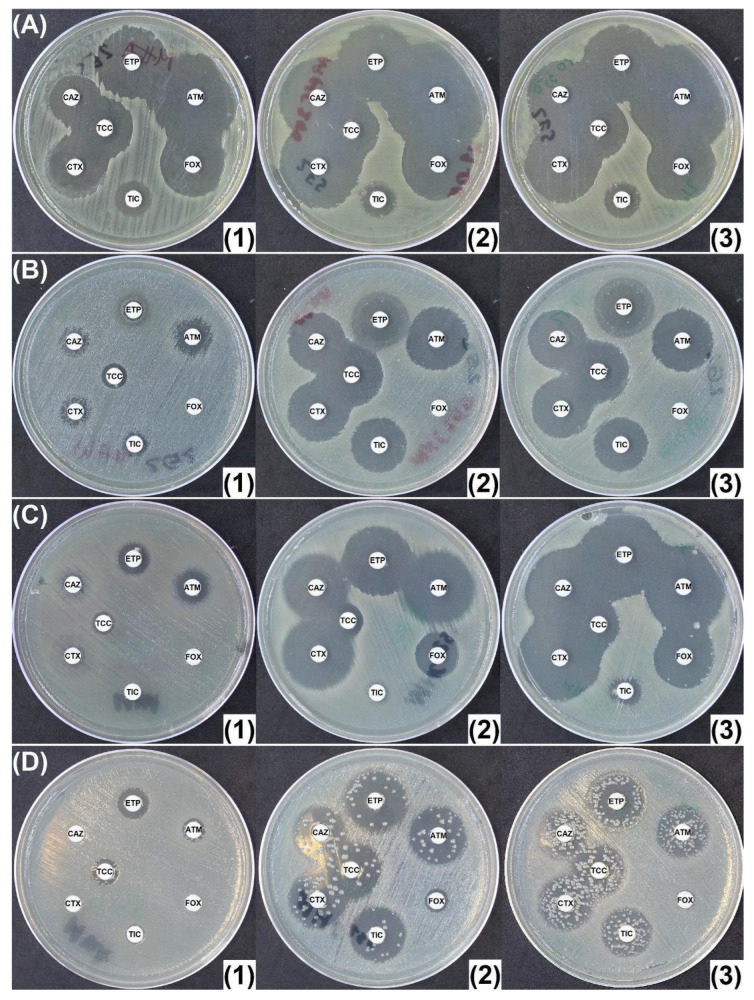
CASE- and HCASE-producing Enterobacterales. (**A**) Cephalosporinase-producing *Proteus mirabilis* (plasmid encoded); (**B**) cephalosporinase-hyperproducing *E. cloacae*; (**C**) cephalosporinase-hyperproducing *E. cloacae*. Larger MHO inhibition zones; (**D**) hyperexpressing mutants that are able to grow even in the presence of inhibitors. (**1**) Mueller-Hinton agar (MHA); (**2**) Mueller-Hinton Agar supplemented with 250 mg/L cloxacillin (MHC250); (**3**) Mueller-Hinton agar supplemented with 300 mg/L oxacillin (MHO300). ATM: aztreonam (30 μg); CAZ: ceftazidime (10 μg); CTX: cefotaxime (5 μg); ETP: ertapenem (10 μg); FOX: cefoxitin (30 μg); TIC: ticarcillin (75 μg); TCC: ticarcillin/clavulanic acid (75/10 μg).

**Figure 6 antibiotics-14-00616-f006:**
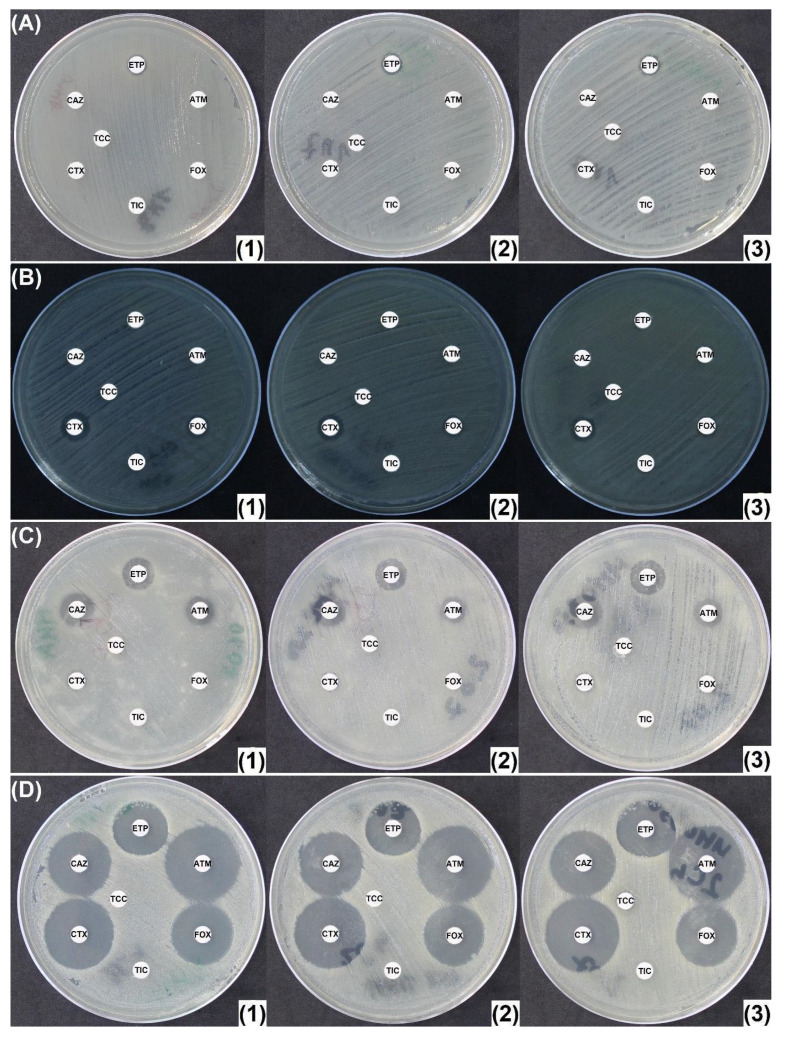
Carbapenemase producers: (**A**) NDM-4 + CTX-M-15 + CMY-6 *E. coli* and high-carbapenem MIC (>32 mg/L for ertapenem, imipenem, and meropenem); (**B**) KPC-2 + SHV-11 *K. pneumoniae*; (**C**) OXA-48 + TEM-1 + CTX-M-15 + OXA-1 *E. cloacae*; (**D**) OXA-181 *E. coli*. The strain presents reduced MICs for cefoxitin, 3rd-gen. cephalosporins, carbapenems, and aztreonam. (**1**) Mueller-Hinton agar (MHA); (**2**) Mueller-Hinton agar supplemented with 250 mg/L cloxacillin (MHC250); (**3**) Mueller-Hinton agar supplemented with 300 mg/L oxacillin (MHO300). ATM: aztreonam (30 μg); CAZ: ceftazidime (10 μg); CTX: cefotaxime (5 μg); ETP: ertapenem (10 μg); FOX: cefoxitin (30 μg); TIC: ticarcillin (75 μg); TCC: ticarcillin/clavulanic acid (75/10 μg).

**Table 1 antibiotics-14-00616-t001:** Statistical tests evaluating differences in inhibition diameters obtained from standard (Mueller-Hinton agar), reference (Mueller-Hinton agar supplemented with 250 mg/L cloxacillin) and investigational media (Mueller-Hinton agar supplemented with 300 mg/L oxacillin). Kruskal–Wallis test by rank is a non-parametric alternative to one-way ANOVA test, to compare between the values of the three groups. The Wilcoxon signed-rank test was conducted to compare the values of MHC250 and MHO300.

	Kruskal–Wallis Test MHA-MHC250-MHO300	Wilcoxon Signed-Rank Test MHC250-MHO300
Aztreonam	0.05355	0.7152
Ceftazidime	0.001271 *	0.05592
Cefotaxime	0.00101 *	0.9244
Cefoxitin	0.006151 *	0.1013
Ticarcillin	0.1358	0.04704 *
Ticarcillin/clavulanic acid	0.001322 *	0.1073
Ertapenem	0.0004103 *	0.5564

* Statistically significant differences (*p* < 0.05).

**Table 2 antibiotics-14-00616-t002:** Characteristics of the bacterial strains used in the study.

No.	Species	Phenotype	β-Lactamase Content	Carbapenemase
1	*K. pneumoniae*	PASE	SHV-28	NA
2	*K. pneumoniae*	BSPASE	TEM-1 + SHV-28	NA
3	*E. coli*	BSPASE	TEM-1B	NA
4	*K. pneumoniae*	ESBL	SHV-11	NA
5	*E. coli*	ESBL	CTX-M-1	NA
6	*E. coli*	ESBL	CTX-M-3	NA
7	*K. pneumoniae*	ESBL	CTX-M-3	NA
8	*E. coli*	ESBL	CTX-M-14	NA
9	*K. pneumoniae*	ESBL	CTX-M-14	NA
10	*E. coli*	ESBL	CTX-M-15	NA
11	*E. coli*	ESBL	CTX-M-15	NA
12	*K. pneumoniae*	ESBL	CTX-M-15	NA
13	*K. pneumoniae*	ESBL	CTX-M-15	NA
14	*K. pneumoniae*	ESBL	CTX-M-15	NA
15	*E. cloacae*	ESBL	VEB-1	NA
16	*E. coli*	ESBL	CTX-M-15	NA
17	*K. pneumoniae*	ESBL	CTX-M-15 + SHV-1	NA
18	*K. pneumoniae*	ESBL	CTX-M-15 + SHV-11	NA
19	*K. pneumoniae*	ESBL	CTX-M-15 + TEM-1 + SHV-11	NA
20	*K. pneumoniae*	ESBL	CTX-M-15 + TEM-1 + SHV-11	NA
21	*K. pneumoniae*	ESBL	CTX-M-15 + TEM-1 + SHV-11	NA
22	*K. pneumoniae*	ESBL	CTX-M-15 + TEM-1 + SHV-12	NA
23	*K. pneumoniae*	ESBL	CTX-M-15 +TEM-1 + SHV-11	NA
24	*K. pneumoniae*	ESBL	CTX-M-15 + TEM-1 + SHV-1 + OXA-1	NA
25	*K. pneumoniae*	ESBL	CTX-M-15, TEM-1B, SHV-28	NA
26	*K. pneumoniae*	ESBL	CTX-M-15, TEM-1B, SHV-83	NA
27	*K. pneumoniae*	ESBL	CTX-M-14	NA
28	*E. coli*	ESBL	CTX-M-2	NA
29	*K. oxytoca*	ESBL	Hyper-K1 (OXY-2-8)	NA
30	*K. pneumoniae*	ESBL + BSPASE	CTX-M-15 + TEM-1 + SHV-1	NA
31	*K. pneumoniae*	ESBL + BSPASE	CTX-M-15 + TEM-1 + SHV-1	NA
32	*K. pneumoniae*	ESBL + BSPASE	CTX-M-15 + SHV-28 + TEM-1	NA
33	*E. coli*	Class D ESBL	OXA-1	NA
34	*K. oxytoca*	Class D ESBL	OXA-2-3	NA
35 *	*E. cloacae*	CASE	CASE	NA
36	*E. cloacae*	CASE	CASE	NA
37	*M. morganii*	CASE	CASE	NA
38	*E. coli*	CASE plasmidic	CMY-2	NA
39 *	*E. cloacae*	CASE + ESBL	CASE + ESBL	NA
40	*E. cloacae*	CASE + ESBL	CASE + CTX-M-15	NA
41	*K. aerogenes*	CASE + ESBL	CASE + TEM-24	NA
42	*E. cloacae*	CASE + Class D ESBL	CASE + OXA-35 + ACT-6	NA
43 *	*E. coli*	HCASE	HCASE	NA
44 *	*E. cloacae*	HCASE	HCASE	NA
45 *	*E. cloacae*	HCASE	HCASE	NA
46 *	*E. cloacae*	HCASE	HCASE	NA
47	*E. cloacae*	HCASE	HCASE	NA
48	*E. cloacae*	HCASE	HCASE	NA
49	*E. cloacae*	HCASE	HCASE	NA
50	*E. cloacae*	HCASE	HCASE	NA
51	*E. cloacae*	HCASE	HCASE	NA
52	*E. cloacae*	HCASE	HCASE	NA
53	*K. aerogenes*	HCASE	HCASE	NA
54	*E. aerogenes*	HCASE	HCASE	NA
55 *	*H. alvei*	HCASE	HCASE	NA
56 *	*S. marcescens*	HCASE	HCASE	NA
57 *	*E. cloacae*	HCASE	HCASE	NA
58	*E. coli*	HCASE plasmidic	DHA-1	NA
59 *	*E. coli*	HCASE plasmidic	ACC-1	NA
60 *	*K. pneumoniae*	HCASE plasmidic	DHA-2	NA
61 *	*P. mirabilis*	HCASE plasmidic	ACC-1	NA
62	*K. pneumoniae*	HCASE plasmidic	SHV + DHA-1	NA
63	*K. pneumoniae*	HCASE + BSPASE	HCASE (CMY-16), TEM-1B, SHV-11	NA
64	*E. coli*	HCASE + ESBL	CTX-M-14	NA
65	*K. pneumoniae*	HCASE + ESBL	LEN16 (99.65%), CTX-M-1	NA
66 *	*E. cloacae*	HCASE + ESBL	HCASE + ESBL	NA
67 *	*E. cloacae*	HCASE + ESBL	HCASE + CTX-M-15	NA
68 *	*E. cloacae*	HCASE + ESBL	HCASE + CTX-M-15	NA
69 *	*E. cloacae*	HCASE + ESBL	HCASE + CTX-M-15	NA
70	*C. freundii*	HCASE + ESBL	HCASE + TEM-3	NA
71	*E. cloacae*	HCASE + ESBL	HCASE + ESBL	NA
72	*E. cloacae*	HCASE + ESBL + BSPASE	HCASE (ACT-7 like), TEM-1B, CTX-M-15	NA
73	*E. coli*	Carbapenemase	NDM-1 + OXA-1 + OXA-10 + CMY-16 + TEM-1	NDM-1
74	*E. coli*	Carbapenemase	NDM-1 + OXA-1 + TEM-1	NDM-1
75	*E. coli*	Carbapenemase	NDM-4 + CTX-M-15 + CMY-6	NDM-4
76	*K. pneumoniae*	Carbapenemase	NDM-1 + OXA-1 + SHV-11	NDM-1
77	*K. pneumoniae*	Carbapenemase	NDM-1 + OXA-1 + CTX-M-15 + TEM-1 + SHV-28 + OXA-9 + CMY-6	NDM-1
78	*P. stuartii*	Carbapenemase	NDM-1 + OXA-1 + CMY-6 + TEM-1	NDM-1
79	*P. rettgeri*	Carbapenemase	NDM-1 + CTX-M-15	NDM-1
80	*K. pneumoniae*	Carbapenemase	VIM-1 + SHV-5	VIM-1
81	*E. cloacae*	Carbapenemase	VIM-1 + SHV-70	VIM-1
82	*E. cloacae*	Carbapenemase	VIM-4 + CTX-M-15 + TEM-1 + SHV-31	VIM-4
83	*K. pneumoniae*	Carbapenemase	IMP-8 + SHV-12	IMP-8
84	*E. cloacae*	Carbapenemase	IMP-8	IMP-8
85	*E. cloacae*	Carbapenemase	GIM-1	GIM-1
86	*E. coli*	Carbapenemase	KPC-2	KPC-2
87	*E. coli*	Carbapenemase	KPC-2 + TEM-1 + OXA-9	KPC-2
88	*K. pneumoniae*	Carbapenemase	KPC-2 + SHV-11	KPC-2
89	*E. cloacae*	Carbapenemase	KPC-2 + TEM-1 + OXA-1	KPC-2
90	*E. cloacae*	Carbapenemase	KPC-2 + TEM-1 + SHV-11	KPC-2
91	*E. cloacae*	Carbapenemase	KPC-2 + TEM-3	KPC-2
92	*E. cloacae*	Carbapenemase	IMI-1	IMI-1
93	*E. asburiae*	Carbapenemase	IMI-2	IMI-2
94	*E. asburiae*	Carbapenemase	IMI-2	IMI-2
95	*E. cloacae*	Carbapenemase	NmcA	NmcA
96	*E. cloacae*	Carbapenemase	GES-5	GES-5
97	*E. cloacae*	Carbapenemase	FRI-1	FRI-1
98	*K. pneumoniae*	Carbapenemase	OXA-48	OXA-48
99	*E. cloacae*	Carbapenemase	OXA-48 + TEM-1 + CTX-M-15 + OXA-1	OXA-48
100	*E. cloacae*	Carbapenemase	OXA-48 + SHV-5	OXA-48
101	*E. coli*	Carbapenemase	OXA-181	OXA-181
102	*K. pneumoniae*	Carbapenemase	OXA-181 + SHV-11 + CTXM-15 + OXA-1	OXA-181
103	*E. coli*	Carbapenemase	OXA-204 + CMY-4 + CTX-M-15 + OXA-1	OXA-204
104	*K. pneumoniae*	Carbapenemase	OXA-204 + SHV-28 + TEM-1 + CTX-M-15	OXA-204
105	*K. pneumoniae*	Carbapenemase	OXA-232 + SHV-1 + TEM-1 + CTX-M-15 + OXA-1	OXA-232
106	*E. coli*	Carbapenemase	OXA-244 + TEM-1 + CMY-2	OXA-244
107	*E. coli*	Carbapenemase	OXA-244 + TEM-1 + CMY-2	OXA-244
108	*P. mirabilis*	Carbapenemase	OXA-23	OXA-23
109	*P. mirabilis*	Carbapenemase	OXA-58	OXA-58
110	*E. cloacae*	Carbapenemase	NmcA	NmcA
111	*E. cloacae*	Carbapenemase	IMI-17	IMI-17
112	*E. cloacae*	Carbapenemase	IMI-17	IMI-17
113	*E. cloacae*	Carbapenemase	IMI-17	IMI-17

PASE = penicillinase; BSPASE = broad-spectrum PASE; ESBL = extended-spectrum β-lactamase; CASE = cephalosporinase; HCASE = hyperproduced cephalosporinase; NA = not applicable. * Isolates used in the preliminary experiment to determine the optimal oxacillin concentration.

## Data Availability

The original contributions presented in this study are included in the article/Appendix A. Further inquiries can be directed to the corresponding author(s).

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
