# Peer review of "Oxacillin-Supplemented Mueller-Hinton Agar for In Vitro Inhibition of Ambler Class C β-Lactamases in Enterobacterales"

_antibiotics, 2025, doi:10.3390/antibiotics14060616_

Round 1
Reviewer 1 Report
Comments and Suggestions for Authors
An interest research

Reviewer 2 Report
Comments and Suggestions for Authors
In this paper, authors evaluated the use of commercially available oxacillin as an alternative to cloxacillin in resource limited countries for efficient inhibition of AmpCs. This study has some merit however I have some minor concern before it is ready for publication.
- Result section 2.1 does not explain the rationale behind selecting MHO300 for this study. Please briefly discuss the results for testing different MHO concentration and selecting MHO300 for further assessment.
- Line 108-109, remove the title for Table S1.
- Remake figure 1 & 2 using different color and shape for different media type for better visualization.
- Similarly, the X and Y legends of Figure 1 & 2 are blurry. Please make it big and clear.
- Why some of the isolated failed to grow on MHC250 or MHO300 plates. Please discuss the potential reason and how low resource labs should handle this? May be using lower antibiotic concentration or increasing broth microdilution.
- Authors did not mention how antibiotics solutions were sterilized before adding to the Mueller-Hinton Agar media. Please explain this.
- Line 33, MHC containing 250 mg/L of what?
- Line 87 “as” is missing after such.
- Line 55-65, please provide the specific citation in the sentences early.
- In introduction, please use consistent β symbol (Line 89 & 99 to line 50 & 57).
- Result 2.1, provide full form of CASE and HCASE when they first appear in the text.
Reviewer 3 Report
Comments and Suggestions for Authors
Please see comments attached. Minor revision is recommended.
